# Detection of Epstein–Barr Virus in Periodontitis: A Review of Methodological Approaches

**DOI:** 10.3390/microorganisms9010072

**Published:** 2020-12-29

**Authors:** Lilit Tonoyan, Marlène Chevalier, Séverine Vincent-Bugnas, Robert Marsault, Alain Doglio

**Affiliations:** 1MICORALIS, Faculté de Chirurgie Dentaire, Université Côte D’Azur, 5 rue du 22ième BCA, 06357 Nice, France; Marlene.CHEVALIER@univ-cotedazur.fr (M.C.); Severine.VINCENT@univ-cotedazur.fr (S.V.-B.); Robert.MARSAULT@univ-cotedazur.fr (R.M.); Alain.DOGLIO@univ-cotedazur.fr (A.D.); 2Pôle Odontologie, Centre Hospitalier Universitaire de Nice, 06000 Nice, France; 3Unité de Thérapie Cellulaire et Génique (UTCG), Centre Hospitalier Universitaire de Nice, 06101 Nice, France

**Keywords:** Epstein–Barr virus, periodontitis, detection methods, EBER-ISH, PCR-based methods, immunohistochemistry, immunophenotyping

## Abstract

Periodontitis, an inflammatory condition that affects the structures surrounding the tooth eventually leading to tooth loss, is one of the two biggest threats to oral health. Beyond oral health, it is associated with systemic diseases and even with cancer risk. Obviously, periodontitis represents a major global health problem with significant social and economic impact. Recently, a new paradigm was proposed in the etiopathogenesis of periodontitis involving a herpesviral–bacterial combination to promote long-term chronic inflammatory disease. Periodontitis as a risk factor for other systemic diseases can also be better explained based on viral–bacterial etiology. Significant efforts have brought numerous advances in revealing the links between periodontitis and Epstein–Barr virus (EBV), a gamma herpesvirus ubiquitous in the adult human population. The strong evidence from these studies may contribute to the advancement of periodontitis research and the ultimate control of the disease. Advancing the periodontitis research will require implementing suitable methods to establish EBV involvement in periodontitis. This review evaluates and summarizes the existing methods that allow the detection and diagnosis of EBV in periodontitis (also applicable in a more general way to other EBV-related diseases), and discusses the feasibility of the application of innovative emerging technologies.

## 1. Introduction

Clinically periodontitis is defined as a chronic multifactorial inflammatory disease characterized by the progressive destruction of the tooth-supporting apparatus. The disease of periodontitis is portrayed by three factors: 1. the loss of periodontal-tissue support manifested through clinical attachment loss and radiographically assessed alveolar bone loss; 2. the presence of periodontal pockets (PP) and 3. gingival bleeding [1]. The periodontal disease initiates as gingivitis (inflammation of the gingiva), which is highly widespread and readily reversible by effective oral hygiene. When left untreated, it may gradually progress to early-to-moderate periodontitis and irreversible advanced periodontitis [2]. Periodontitis, along with dental caries, is considered one of the two biggest global oral health burdens [3]. Beyond oral health, growing evidence in the literature supports the direct and indirect impact of periodontitis on the overall health and development of extraoral pathologies. Periodontitis has been associated with seemingly unrelated systemic diseases such as diabetes, cardiovascular diseases and stroke, adverse pregnancy outcomes, respiratory diseases, dementia, Alzheimer’s disease, rheumatoid arthritis and different types of cancers [4,5]. However, it remains to be further scrutinized how specific periodontal pathogens contribute to the development of systemic diseases. However, as a minimum, the detection of periodontal pathogens in the oral cavity may be used as an attractive tool for the diagnosis of non-oral inflammatory systemic diseases.

There is a long history of the search for etiological agents of periodontitis and different hypotheses of etiopathogenesis have been proposed. Periodontitis was thought to be (i) an infection caused by bacteria; (ii) a specific bacterial infection; (iii) a biofilm infection; (iv) a specific plaque; (v) result from dysbiosis; (vi) caused by complex interactions among bacteria–host–environmental factors and (vii) a viral-bacterial infection (reviewed in [5,6]). One specific recent hypothesis of interest is based on the herpesvirus–pathogenic bacteria–host response axis in which herpesviral–bacterial interactions assume a major etiopathogenic role [5,7,8,9]. This infectious disease model for periodontitis development proposes that bacteria initiate the gingival inflammation triggering further influx and propagation of herpesviruses. Next, a herpesvirus active infection in the periodontium hinders the local immune defenses, thereby permitting the overgrowth of periodontopathic bacteria. In a two-way interaction, the virulence factors of periodontopathic bacteria reactivate latent herpesviruses and augment the infection. Reactivated periodontopathic herpesviruses and bacteria also modulate host immune reactions and provoke tissue destruction as a result of immunopathologic responses leading to the progression of the disease. In particular, among herpesviruses, human cytomegalovirus (HCMV) and Epstein–Barr virus (EBV) have been closely associated with severe types of periodontitis [10]. The main focus of the discussion in this review is EBV.

EBV belongs to the family of human gamma herpesviruses (systematic name human herpesvirus 4—HHV-4) and is one of the most ubiquitous and successfully adapted human pathogens that are found in approximately 95% of the total human population. EBV can infect a wide variety of cells and tissues, mostly B cells, nasopharynx and oropharynx squamous epithelial cells (ECs), thyroid glandular ECs, salivary and stomach glands and, occasionally, T cells, smooth muscle cells and follicular dendritic cells [11]. EBV has been associated with an extended list of diseases, from transient benign infections to aggressive malignancies. EBV is best known as the causative agent of infectious mononucleosis and has been implicated in several oral pathologies, such as oral hairy leukoplakia (OHL), oral lichen planus, Sjogren’s syndrome and periodontitis [9]. It is a known carcinogen implicated in the etiology of several malignancies of both lymphoid and epithelial origin [12]. Infection of B cells with EBV has been linked to Burkitt’s lymphoma, Hodgkin lymphoma and post-transplant lymphoproliferative disorders; infection of ECs is implicated in nasopharyngeal cancer, gastric cancer and breast cancer. Furthermore, a recent study suggested the association of EBV with seven different autoimmune diseases—multiple sclerosis, rheumatoid arthritis, inflammatory bowel disease, type 1 diabetes, juvenile idiopathic arthritis and celiac disease [13]. To confirm the etiopathogenic role of EBV in a disease, a better understanding of the EBV biology and molecular mechanism of the associated disease is required.

EBV is an enveloped DNA virus that has approximately 172 kb double-stranded DNA (dsDNA) genome encoding genes for latent and lytic infection. This virus was the first of the herpesviruses to be completely sequenced [14] to identify over 80 protein-coding open reading frames and around 30 different non-coding RNAs (ncRNAs) [15].

Latent EBV infection allows for long-term viral persistence in the host, owing to tight control of viral gene expression to reduce the antiviral immune recognition. During latent infection, only several different types of RNAs and proteins are expressed. They include ncRNAs (EBV-encoded RNA 1 (EBER1) and EBER2, microRNAs (miRNA), EBV-stable intronic-sequence RNAs (EBV-sisRNA), EBV small nucleolar RNAs (EBV-snoRNA) and RPMS1 messenger RNA), six nuclear proteins (EBV nuclear antigen 1 (EBNA1), EBNA2, EBNA3A, EBNA3B, EBNA3C and EBNA5) and three latent membrane proteins (LMP1 and LMP2A-B) [11,15].

Upon lytic reactivation, EBV genes are sequentially expressed in immediate-early (IE), early (E) and late (L) states. Switch from latent to lytic state is triggered by expression of two IE viral transcription factors, the master regulator ZEBRA (also known as BZLF1, Zta, EB1 or Z) and Rta (BRLF1 or R). ZEBRA and Rta individually or cooperatively activate a subset of E genes many of which encode proteins involved in viral lytic DNA replication, such as the single-stranded DNA-binding protein (BALF2) and five replication enzymes and coenzymes, namely the helicase (BBLF4), primase (BSLF1), primase-associated factor (BBLF2/3), DNA polymerase (BALF5) and DNA polymerase processivity factor (BMRF1) [16]. For more details of E gene products, the reader is referred to the discussion in Kenney, 2007 [17]. Viral DNA replication is followed by expression of EBV’s L genes, which code for viral structural proteins (major capsid protein p160 (BcLF1) and three small capsid proteins, p18, p23 and p40 (BFRF3, BLRF2 and BdRF1)), glycoproteins (gp350/220 (BLLF1), gH (gp85; BXLF2), gp42 (BZLF2), etc.) and tegument proteins and viral interleukin 10 (vIL-10; BCLF1) [16,17,18]. All these latent and lytic determinants (the whole list is out of the scope of this review) are potentially useful for EBV detection and objective diagnosis.

Accurate laboratory tests to detect EBV are needed for purposes of basic and epidemiologic research and clinical management for different diseases. Biochemical, serological, immunological, histological, cytological and molecular detection methods of EBV have been used in the diagnosis and monitoring of patients with EBV-associated diseases [19]. The development of advanced laboratory methods allows timely and accurate diagnosis of clinical manifestations, which, in turn, may contribute to the prognosis and successful treatment. The identification of a suitable methodology that links EBV with different diseases will also advance our understanding of the molecular mechanisms underlying the onset and progression of EBV-associated diseases. The main aim of the present review is to compile EBV detection and quantification methods with a focus on periodontitis (outlined in Figure 1). Although this review represents a practical guide for the periodontitis model, it may also pave the way for the understanding of other inflammatory infectious diseases associated with EBV infection.

## 2. Sampling

The physical presence of EBV in periodontal lesions suggests that EBV may be implicated in the etiopathogenesis of periodontitis. As such, samples need to be taken from the periodontal environment. The sample types and sample extraction methods may influence the identification and enumeration of microbes. The most eligible articles studying the association of EBV with periodontitis used subgingival plaque (SbgP), gingival crevicular fluid (GCF) and tissue/biopsy samples as sample type/sampling location [20], and curette, paper point, paper strip and surgery/biopsy as sample-extracting methods [21]. There is also a good deal of literature retrieving EBV from peripheral blood and saliva of periodontitis patients. Considering EBV is ubiquitous in the human population, the blood will not be reviewed here as a sample relevant to periodontitis, consequently, the serological tests are not discussed in this review. Though salivary EBV load may be very relevant to periodontitis, in this review, the saliva as a sample and salivary EBV load detection methods are not discussed either, because EBV is ubiquitous in humans, it is transmitted through saliva and EBV DNA is commonly detected also in the saliva from healthy adults [22].

The inspection of the subgingival plaque occupying the PP is considered the gold standard in studying periodontitis-associated microbial communities [23]. As might be expected, site-specific, intra- and interindividual variations of SbgP profiles may occur. In general, several paired SbgP samples are collected from shallow (healthy) and deep (diseased) sites from the same patient. The single-site analysis is preferential, but for practical and economic reasons, pooled SbgP samplings have often been performed [23]. Paper points are widely used for the collection of SbgP. Generally, the area of the collection is isolated with cotton rolls and air-dried to avoid contamination with saliva, then the supragingival plaque and calculus are carefully removed with a scaler to ensure the collection of only the subgingival material [24]. The color-coded paper points of specific sizes are inserted into the base of the PP, left in place for a certain duration and eluted. Basic parameters such as the origin of the paper points (manufacturer/supplier), the ISO size, probing (sampling) time and elution time may influence the optimum conditions for the microbiological sampling of PPs [25]. Samples can be collected as single (one paper point into the PP of each tooth), pooled (several paper points into the PPs of several teeth) or parallel (several paper points at one tooth) samples [26]. 

Curettes are also commonly used for sampling of subgingival specimens. After isolating the area with cotton rolls, a sterile curette tip is gently introduced through the pocket orifice into the bottom of the pocket and then removed with slight pressure against the tooth in a single vertical stroke to obtain the subgingival material [27,28].

In periodontitis, gingival crevicular fluid is an inflammatory exudate comprised of host-related substances, and from supra- and subgingival located microbes, thus, the analysis of GCF has become more and more important in the diagnosis of periodontitis [29]. Paper strips are used for GCF sample collection. The paper strip is inserted into the gingival crevice (intracrevicular method) or overlaid on the gingival crevice region (extracrevicular method). The intracrevicular method is subdivided to (i) superficial, when the strip is inserted just at the entrance of the crevice or PP and (ii) deep, when the strip is inserted to the base of the pocket or until minimum resistance is felt [30].

Tissue/biopsy specimens are obtained by periodontal surgery, which contain gingival tissue located adjacent to the PP. Careful dissection of the surgical piece can enrich the biopsy specimen with periodontal tissue attached to the tooth while removing the more distant gingival conjunctive areas. Dependent on the EBV detection method, the tissue specimens can further undergo cell dissociation, homogenization or fixation for isolation of macromolecules or histological analyses.

## 3. Polymerase Chain Reaction (PCR)-Based Detection Methods

PCR-based detection and quantification of EBV nucleic acids in body fluids and tissues have been used in the diagnosis and monitoring of EBV-associated diseases [19]. Extensive literature exists describing the application of PCR-based methods to identify and quantify EBV in periodontitis, which has targeted different genomic regions of EBV and applied different types of PCR methods. While most studies refer to the detection of the viral genomic DNA suitable to estimate the amounts of viruses in a specimen, a few other studies have focused on the detection of viral transcripts that can be more related to stages of viral replication in infected tissues. Specific characteristics and outcomes of recent studies (2010–2020) are summarized in Table 1. The large majority of these studies ascertain a strong association of EBV with periodontitis and its severity, indicating that EBV may serve as an etiopathogenic factor in periodontal diseases.

Conventional singleplex PCR was applied to amplify and detect a single target gene of EBV, such as EBNA2 [31] and LMP2 [32], while multiplex PCR was used to simultaneously amplify target sequences of several periodontopathic bacteria and herpesviruses [33,34,35,36]. Another PCR technique, the nested PCR, was implemented to increase the specificity of EBV DNA amplification and reduce the non-specific amplification by the involvement of two sets of primers (outer and inner pairs) for the same target [37,38,39,40,41]. Additionally, the nested PCR is more efficient in detecting low viral loads [42]. Typically, the amplicons generated via these PCR methods are subsequently size-fractioned and detected by agarose (AGE) or polyacrylamide gel electrophoresis (PAGE) (Table 1), or else, restriction fragment length polymorphism analysis (RFLP) with endonucleases followed by AGE can be applied. For example, Afa I digests the 497 bp amplicon of EBV1 in 355 bp and 142 bp fragments, while Stu I digests the 165 bp amplicon of EBV2 in 118 bp and 47 bp fragments, which can be visualized with AGE [43].

Due to EBV ubiquity and life-long persistent infection, simply detecting it is insufficient to diagnose EBV association with the disease [44]. Quantitative measurement of the EBV genome is necessary to distinguish between low-level EBV infection in healthy carriers and high levels typical to EBV-associated diseases [45]. Real-time quantitative PCR (further named qPCR) is the main method for modern EBV viral load measurement, which also eliminates post-PCR manipulations. Real-time qPCR is based on the amplification of a conserved sequence (typically around 100 bp) using either a fluorescent probe (e.g., TaqMan probe and fluorescence resonance energy transfer (FRET) hybridization probe) or an intercalating dye (e.g., SYBR Green) coupled with real-time laser scanning to quantify the target DNA against serial dilutions of known EBV DNA content [11,45]. Quantified EBV genomic DNA (gDNA) sources are now commercially available that may be used for assay calibration [45].

In the TaqMan system, in addition to the two amplification primers used in conventional PCR, a dual-labeled fluorogenic hybridization probe is used. The probe hybridizes specifically in the DNA target region between the two PCR primers. One fluorescent dye serves as a reporter and its emission is quenched by the second fluorescent dye. Nuclease degradation of the hybridization probe by Taq DNA polymerase releases quenching of the reporter fluorescence, resulting in an increase in peak fluorescence [44,46]. The principle of FRET hybridization is based on the hybridization of two single-stranded, sequence-specific, fluorescent-labeled oligonucleotides (with donor and acceptor dyes) to the target sequence in close proximity in a head-to-tail orientation. The energy absorbed by the donor fluorophore is transferred to the acceptor fluorophore, which then emits fluorescence (FRET) [47]. In the SYBR Green system, a green dye is used as a marker for product accumulation, which intercalates into dsDNA as PCR products accumulate [45,48]. The SYBR Green system is less expensive but less specific in comparison with probe strategies.

Real-time qPCR is considered a sensitive, reliable, stringent, simple, specific, precise and fast method [11]. Since nucleic acid amplification and detection occurs in the same sealed tube, the risk of amplicon contamination is negligent compared with conventional PCR methods. Due to advanced instrumentation, the real-time qPCR testing is much simpler to perform and the test results are acquired much faster [49]. Currently, several real-time qPCR-based EBV detection and quantification kits are commercially available, such as EBV R-GENE (bioMérieux), EBV ELITe MGB (ELITechGroup), artus EBV PCR Kits (QIAGEN), etc. However, attention should be paid when comparing the data of different studies [50]. The source of deviation could be the units of measurement (copies per milliliter, copies per microgram of DNA, copies per positive cell) or the EBV targets (LMP2, BKRF1 or BamHI W (EBNA1), BNRF1 (membrane protein), BXLF1 (thymidine kinase), BZLF1, BALF5 or BHRF1 (transmembrane protein), etc.).

EBV DNA can be detected with high specificity, sensitivity and rapidity on par with the real-time qPCR method utilizing the loop-mediated isothermal amplification (LAMP) method. The LAMP reaction requires a DNA polymerase with strand displacement activity and a set of four specially designed inner and outer primers that recognize a total of six distinct sequences within the target DNA. Iwata et al. (2006) designed primers for the EBV LAMP assay based on BamHI W gene sequences [51]; Liu et al. (2013) later designed an extended set of LAMP primers for latent (EBNA1, EBNA2, LMP1 and LMP2A) and lytic (BZLF1) transcripts [52]. During LAMP reaction specific DNA targets are amplified at 63–65 °C, without thermocycling, accumulating 10^9^ copies of the target in less than an hour. The final products of LAMP are stem-loop DNAs with several inverted repeats of the target DNA and cauliflower-like structures with multiple loops. LAMP amplicon product is further detected by turbidity assay (TA) of the white precipitate of magnesium pyrophosphate and/or AGE. The reaction is described in detail in [51,53]. Elamin et al. used this technique to assess the presence of putative periodontopathic bacteria (*Aggregatibacter actinomycetemcomitans*, *Porphyromonas gingivalis*, *Tannerella forsythia* and *Treponema denticola*) and two periodontal herpesviruses (EBV1 and HCMV) in individuals with aggressive periodontitis [54]. Though they reported no significant association between EBV1 and the disease, the highest risk of aggressive periodontitis was observed when *A. actinomycetemcomitans* was detected together with EBV1 and/or HCMV.

Using a sensitive and reproducible reverse transcription qPCR (RT-qPCR) method different transcripts of EBV can be detected and quantified to distinguish distinct states of latent or lytic EBV infection or closely monitor reactivation of EBV. In theory, RT-qPCR differs from qPCR only by the addition of a preliminary step, the initial complementary DNA (cDNA) synthesis from an RNA template by an RNA-dependent DNA polymerase (reverse transcriptase). After the RT reaction, suitable detection chemistry to report the presence of PCR amplicons, an instrument to monitor the amplification in real-time and appropriate software for quantitative analysis are required [55]. From the list of recent studies (Table 1), Hernádi et al. used the RT reaction to convert the EBNA2 messenger RNA into cDNA followed by nested PCR to detect that EBNA2 expression was significantly more frequent in apical periodontitis lesions as compared to healthy controls [38]. They concluded that EBV infection was a frequent event in apical periodontitis and that symptoms were likely to occur if the lesion is aggravated with active EBV infection. Vincent-Bugnas and coauthors used the sensitive RT-qPCR technique and observed that EBV latent (EBNA1, EBNA2, LMP1 and LMP2) transcripts were detectable in all PP samples of chronic periodontitis (CP) patients, which were within the range expressed by EBV-infected cell lines [27]. EBNA1 was expressed at the highest and very similar levels to those measured in EBV-infected cell lines. Moreover, the EBNA1 expression level was correlated with the severity of the CP. On the other side, the IE viral transactivator BZLF1, known to induce the EBV lytic cycle, was also expressed in CP samples but at a level lower than that observed in the EBV-producing cell line. Overall, their conclusions derived from RT-qPCR analysis were that EBV-infected periodontal cells were likely in a state of latent EBV infection and that the level of EBV infection correlated with disease severity.

## 4. Immunohistochemistry (IHC)

To identify the precise cellular location of EBV, morphology-based techniques are used. IHC may be applied to confirm the presence, distribution, localization of EBV in the cells/tissues and distinguish latent from lytic infection based on protein expression profiles. IHC for EBV detection involves the staining of key EBV proteins such as EBNA1, EBNA2, LMP1, LMP2A and BZLF1 [63]. Commercial antibodies to EBV for IHC assays are available. There are also automated and standardized procedures routinely and widely used in pathology laboratories to detect EBV proteins in tissue specimens, such as FLEX monoclonal mouse anti-Epstein–Barr virus, LMP, Clones CS.1–4 (DAKO), which are used together with Autostainer Link instruments. 

IHC procedures are performed on formalin-fixed, paraffin-embedded (FFPE) tissue sections of periodontal biopsies and cytological preparations from the periodontal environment. A standard IHC protocol is a multistep procedure involving deparaffinization/rehydration, heat- or proteolytic-induced antigen retrieval, blocking of non-specific staining, permeabilization, immunostaining with a primary antibody specific to a target antigen, incubation with labeled secondary antibody and detection. IHC allows for chromogenic (chromogenic immunohistochemistry—CIHC) and fluorescent (immunofluorescent—IF) detection types. For CIHC detection, the antibody is conjugated to an enzyme (such as horseradish peroxidase (HRP) or alkaline phosphatase (AP)), which converts a substrate (such as 3,3′-diaminobenzidine (DAB) or 3-amino-9-ethylcarbazole (AEC)) into a colored precipitate at the antigen site. For IF detection, the fluorophore (such as Alexa Fluor family dyes or fluorescein isothiocyanate (FITC)) conjugated antibody is excited by and emits light at specific wavelengths. Following the immunostaining, counterstaining with hematoxylin (chromogenic detection) or with DAPI (fluorescence detection) is performed to contextualize the antigen of interest. After the completion of all staining, the tissue is mounted and visualized by a bright-field (CIHC) or fluorescence/confocal (IF) microscope.

Multiplex chromogenic and fluorescence immunohistochemistry has recently emerged as a potent tool for the simultaneous detection of multiple biological markers on a single tissue section using a consecutive or simultaneous staining approach [64]. Multiplexed strategies allow compiling maximal information per tissue section of a limited sample and to understand coexpression and colocalization of multiple targets within tissue architecture.

Though IHC is a sensitive, versatile technique with many applications, careful control selection and proper optimization of the protocol is required. Besides, because the evaluation of the staining intensity of IHC is subjective, the ambiguity in the evaluation of the results and inter- or intraobserver variability may be problematic [65].

IHC, multiplex IHC and combined IHC techniques were employed for EBV analysis in periodontitis research (Table 2). In this context, using the CIHC approach for LMP1 protein immunostaining Saboia-Dantas et al. observed EBV in 31% of apical periodontitis lesions obtained after teeth extraction [66]. Vincent-Bugnas et al. applied IF costaining of viral latent proteins LMP1 and LMP2, and junctional EC marker cytokeratin 19 (CK19) to detect latent EBV-infected periodontal ECs (pECs) in non-surgical liquid-based cytological samples derived from PPs of CP patients (Figure 2 [27]). They estimated that around 32% of the CK19^+^ cells were infected with EBV (LMP2^+^).

## 5. In Situ Hybridization (ISH)

EBV RNA and DNA can be readily localized in specific cells or tissues with ISH, which combines molecular biological techniques with histological and cytological analysis of gene expression.

ISH to EBV DNA: To localize a virus in the tissue samples ISH for the detection of EBV DNA can be used. Most of the time, probes target the BamHI W fragment of the EBV genome, which is repeated up to 15 times in the EBV genome [69]. This approach is more sensitive than the use of single-copy gene probes. However, it is less practical to target EBV DNA rather than EBER RNA for ISH (discussed below), unless the RNA in the sample has been selectively degraded [63].

EBER in situ hybridization (EBER-ISH): EBER-ISH is often regarded as the gold standard technique to detect EBV in human specimens. EBERs represent reliable molecular targets to detect and localize EBV-infected cells in tissue samples as EBERs are ubiquitously expressed in all known latency states at levels greater than 10^6^ copies per infected cell [70]. However, EBER transcripts may lack in exclusively lytic infections, such as OHL in human immunodeficiency virus (HIV)-infected patients [71]. EBER1 and EBER2 are non-polyadenylated RNAs (167 and 172 bp, respectively) located in the nucleus, which are actively transcribed by RNA polymerase III but remain untranslated [72].

EBER-ISH can be performed in several ways using oligonucleotide DNA probes, RNA probes (riboprobe) or peptide nucleic acid (PNA) probes labeled with radioactive tags, biotin, digoxigenin or fluorescein [63]. There are a handful of different commercially available EBER probes and detection kits. A basic EBER-ISH workflow consists of several interdependent steps and is similar to ICH staining. It is employed on tissue sections or cytological samples, which are formalin-fixed, dehydrated, paraffin-embedded and sectioned by routine methods. The procedure starts with deparaffinization in xylene and rehydration through a series of graded ethanols. The specimens are further pretreated with proteinase K to enhance probe entry into the nucleus where EBER transcripts are situated. The characteristic step for EBER-ISH is that the labeled probe is hybridized to the target EBERs at elevated temperature and the excess unbound probe is washed away. Subsequently, the detection of the labeled probe and counterstaining are performed. Interpretation of EBER^+^ signal depends on microscopic visualization of the nuclear EBER staining in infected cells.

Though EBER-ISH is considered the gold standard for the evaluation of EBV-positivity there are still pitfalls in the technique and interpretation of results [73]. False-positive results can be related to latent infection of background lymphocytes, non-specific staining or cross-reactivity with mucin. False-negative results may be related to RNA degradation. To overcome the latter problem, control hybridization for a ubiquitous cellular transcript must be run in parallel to ensure that RNA is preserved and available for probe binding in the cells of interest. The U6 cellular transcript (non-coding small nuclear RNA) is a suitable control due to its similarity to EBER in terms of size, abundance and intranuclear localization [63]. In addition, the technique is only applicable to infected cells and does not detect the cell-free virus [44].

Multilabeling techniques: To assure a correct assignment of EBV infection to a specific cell type multilabeling techniques may require simultaneous detection of viral RNA or viral DNA or viral gene products on the one hand and cell-specific markers on the other hand [69]. Multilabeling sequentially performed on the same tissue section slide will also reduce the consumption of limited tissue [74,75]. Thus, EBER-ISH is often used in combination with other techniques, such as IF and CIHC, to simultaneously detect EBERs and cellular markers (Table 2).

Several studies utilized EBER-ISH to detect EBV in periodontitis lesions (Table 2). Vincent-Bugnas and coauthors employed EBER-ISH in combination with CIHC for CK19 immunostaining and observed notable EBV infection in pECs of PP samples from CP patients (Figure 3a,b [27]). This EBER-ISH/CIHC analysis showed that the frequency of EBV^+^ pECs was higher in deep pockets than in shallow pockets. Overall, analyzing the EBER^+^ pEC frequency measured in CP patients and healthy donors, they found a positive correlation with the level of disease progression. In addition, the combo of EBER-ISH with CIHC for chemokine ligand 20 (CCL20) staining showed that 80% of EBER^+^ pECs from CP (Figure 3c) but not from healthy samples were also positive for CCL20. Considering CCL20 is a pivotal inflammatory chemokine that controls tissue infiltration by immune cells, they hypothesized that EBV could worsen the local inflammatory state by promoting the production of CCL20. Kato et al. combined EBER-ISH with CIHC staining of CD19 (cluster of differentiation; B cell marker) and showed that B cells abundantly infiltrated into the gingival connective tissues subjacent to the gingival epithelium in periodontitis patients [39]. Interestingly, in the same location, a large number of B cells were EBER^+^. The study from Makino et al. employed EBER-ISH to detect EBER in the cytoplasm and nuclei of B cells and plasma cells (PC) in 66.7% of periapical granulomas, but not in healthy gingival tissues [68]. It should be mentioned that the periapical granulomas are formed due to the chronic inflammation caused by periapical periodontitis, which are comprised of granulomatous tissue embedded with inflammatory cells such as polymorphonuclear leukocytes, lymphocytes, PCs and macrophages. In addition, in the same study, the IHC of latent membrane protein LMP1 showed that all of the EBER^+^ periapical granulomas were also positive for LMP1, and EBER-expressing cells were localized in the same areas as LMP1-expressing cells. In a very recent study, Olivieri and coworkers used multiplex IHC staining and detected T cells (CD3 marker), B cells (CD20 marker), PCs (CD138 marker) and antibody-producing cells (cytoplasmic kappa light chain) on paraffin-embedded serial sections from biopsies of periodontitis patients (Figure 4a–c [62]). On the same serial sections, EBER-ISH revealed numerous EBV-infected cells whose localization matched that of CD138^+^ PCs. Interestingly, Sunde et al. using EBER-ISH could not detect cells harboring EBV in apical samples obtained by submarginal incision from periodontitis patients [67].

## 6. Immunophenotyping

While microscopy is used to visualize cells based on their morphology and staining characteristics, using flow cytometry (FCM) cells can be “visualized” qualitatively and quantitatively based on similar characteristics. As such, FCM immunophenotyping can be employed to address the topic of EBV infection of periodontal cells.

FCM is a widely used method to analyze the expression of cell surface and intracellular molecules, characterize different cell subpopulations in a complex cell population, and analyze the size, internal complexity and volume of the cells, allowing simultaneous multiparameter analysis of single cells. FCM measures the fluorescent intensity produced by fluorochrome-conjugated monoclonal antibodies (mAb) that have a specific affinity for an antigen on the cell surface, a protein in the cytoplasm or the DNA in the nucleus [76]. Many cell surface and some internal antigens can be simultaneously assessed by employing different combinations of fluorochromes and conduct multicolor (multiparameter) experiments. Current advances allow the measurement of as many as 30 fluorescent parameters simultaneously [77]. An excellent collection of broad guidelines for the use of FCM in immunology was recently published by the editorial team of the European Journal of Immunology [77].

FCM requires single dissociated cells in a liquid medium. If cells from a solid tissue (such as a periodontal biopsy) have to be analyzed, a disintegration of the tissue into single cells is imperative for the flow analysis [77]. Two main dissociation techniques are enzymatic digestion and mechanical disintegration or the combination to give the best results. The dissociation is followed by the cell number and viability determination after which the cell suspension can be used directly for FCM analysis or stored after fixation or freezing for later measurement.

Although FCM holds a promise to provide useful diagnostic information in EBV-related diseases, the literature lacks articles reporting on the use of FCM in studying the EBV–periodontitis relationship. In practice, reliable EBV-specific labeled antibodies for FCM are somewhat non-existent, so the FCS is usually combined with EBER-ISH. As such, cell suspensions can be stained for different cell markers, followed by fixation and EBER-ISH after which can be used for FCM analysis, allowing the quantification of EBV-infected cells and simultaneous characterization of the infected cell phenotype [78,79].

Further study of different cell types and subsets often requires the isolation of specific populations that can be realized by a technique known as fluorescence-activated cell sorting (FACS). Flow cytometers with sorting capabilities can detect cells by size, morphology and protein expression and then using droplet technology sort cells and recover the subsets for post-experimental use. These cells can be further lysed to interrogate nucleic acid, protein or metabolite content. Considering there is a lack of reliable commercial anti-EBV mAbs directly linked to fluorescent dyes for FACS analysis, cell populations that are suspected of harboring EBV, such as B cells, PCs, T cells and ECs, can be sorted based on their specific surface markers and discrimination of non-specific markers directly into the lysis buffer for downstream EBV detection, for example, using a nucleic acid amplification method. As mentioned, there is no available literature reporting the direct usage of FCM/FACS in EBV detection from periodontal material. In a recent article [62] the authors used the multiparameter FACS analysis of immune infiltrate derived from the PP biopsies of severe periodontitis patients and observed that the majority of the immune cells were CD3^+^ T cells, while among the B cell subpopulations, plasmablasts (CD10^−^, CD20^−^, CD38^+^) and a lesser extent immature B cells (CD10^+^, CD20^+^, CD38^−^) were prominent. However, the presence of EBV the authors could only confirm using multilabeling techniques—EBER-ISH coupled with multiplex CIHC (described in the multilabeling techniques section; Figure 4) performed on the small tissue samples originated from the same biopsies as for FACS analysis.

Besides FACS, other cell sorting and enrichment methods exist, such as magnetic-activated cell sorting (MACS) when magnetic bead-conjugated antibodies target specific cell surface molecules [80]. Based on antibody–antigen interactions, magnetic beads-coated cells are separated from non-coated cells when passing through a magnetic field. MACS can be achieved with positive (recognize and conjugate the target cell) or negative (recognize the unwanted cells) separation. MACS technology products are mainly manufactured by Miltenyi Biotec, also by Sepmag Technologies, TurboBeads, STEMCELL Technologies, Thermo Fisher Scientific, etc. [77]. Though MACS is considered as a flexible, fast, specific and simple cell sorting system, only a limited number of molecules can be targeted, as magnetic beads are indistinguishable, unlike various fluorescent colors in FACS. In addition, unintended activation of target cells by bead attachment is possible [77,80]. Contreras et al. used the MACS technique to isolate polymorphonuclear neutrophils (CD15^+^), monocytes/macrophages (CD14^+^), T cells (CD3^+^) and B cells (CD19^+^) from biopsy-derived cell fractions of periodontitis patients [81]. After MACS cell sorting, the DNA was extracted and EBV1 and EBV2 were detected by nested PCR and AGE for EBNA2 gene identification. They observed that EBV1 was present in 50% of tissue biopsy specimens and 45% of B cell fractions from the adult periodontitis lesions.

## 7. Advanced Prospective Methods

Several novel technologies hold great potential to be implemented in the detection and study of EBV in periodontitis.

### 7.1. PrimeFlow RNA Assay

As discussed prior, reliable and consistent methods for intracellular staining of distinct EBV-infected periodontal cell populations for evaluation by FCM have been problematic. The limitation of direct access to EBV-infected cells hampers progress in the understanding of mechanisms underlying the etiopathogenesis of the periodontal disease. A recently developed ISH technique, the PrimeFlow RNA assay (Thermo Fisher Scientific), combines the power of the branched DNA (bDNA) technology with the single-cell resolution of FCM. This assay provides simultaneous detection of up to four RNA targets in combination with immunophenotyping for cell surface and intracellular markers using fluorochrome-tagged antibodies allowing further differentiation of specific cell subpopulations.

The assay workflow includes several steps (described more detailed by the manufacturer): preparation of the suspension cell sample, antibody staining, fixation and permeabilization (including intracellular antibody staining, if desired) and target hybridization with target-specific probe pairs. The subsequent signal amplification step of the workflow includes a series of sequential hybridization steps of the target probe to bDNA structures formed by preamplifier, amplifier and label probes. bDNA technology amplifies the detection of an RNA transcript, rather than the target RNA itself. A fully assembled signal amplification “tree” has 400 label probe binding sites providing 8000–16,000-fold amplification. At the detection step, the fluorochrome-conjugated label probes allow the detection of the target using a standard flow cytometer.

Oko and coauthors concluded that this technique affords new opportunities to understand the complexity of virus infection within single cells when they observed that EBV infection (gamma herpesvirus infection in general) can be surprisingly heterogeneous at the level of the individual cell [82]. Using the PrimeFlow assay, very recently Fournier et al. showed that during chronic active EBV infection, EBV-infected B cells from blood expressed PC differentiation markers [83]. Taking into consideration these achievements, the PrimeFlow assay may also be successful in EBV–periodontitis research.

### 7.2. RNAscope Assay

RNAscope technology is a novel RNA-ISH assay for the detection of target RNA within intact cells. Single-molecule RNA visualization in individual cells is achieved by the use of a novel probe design strategy and a hybridization-based signal amplification system to simultaneously amplify target-specific signals and suppress background noise (detailed description in [84]). The steps in RNAscope are similar to those in IHC and PrimeFlow RNA assay. Initially, tissue sections or cells are fixed onto slides and pretreated to unmask target RNA and permeabilize cells. The permeabilization is followed by the hybridization of target-specific double oligonucleotide probes (ZZ) to multiple RNA targets. Sequential hybridizations with the preamplifier, amplifier and label probe can theoretically yield up to 8000 labels for each target RNA molecule. Each unique label probe is conjugated to a different fluorophore or enzyme. In the visualization step, signals are detected using an epifluorescent microscope (for fluorescent label) or standard bright-field microscope (for enzyme label). In the final quantification step, single-molecule signals can be quantified on a cell-by-cell basis by manual counting or automated image analysis using HALO Software. Importantly, this technique allows multiplex detection of RNA and protein targets simultaneously [85], as well as multiple mRNA species with different fluorophores can be resolved [86], such as latent (EBER1, EBNA1, LMP1 and RPMS1) and lytic (BZLF1 and BMRF1) targets [87]. This highly sensitive and specific RNAscope assay may allow new insights into viral reservoir, persistence and evaluation of treatment strategies [85].

### 7.3. QIAScout Microraft Array

Single-cell analysis retains a wealth of information that is lost when studying instead bulk population of cells. Recent advances (other than FACS and MACS) have enabled the precise isolation of selected single cells from complex samples. Recently, QIAscout (QIAGEN) has been developed to effectively and rapidly isolate and recover the single cell by microrafts. The cell remains viable after the separation and can be used for downstream analysis such as PCR, next-generation sequencing (NGS), whole-genome sequencing (WGS), whole-transcriptome sequencing (WTS) or for clonal expansion. QIAScout technique requires a single-cell suspension, which is placed in the microraft array and the cells are allowed to settle. The microrafts serve as releasable culture “nests” for individual cells. QIAscout arrays are mounted on a standard inverted microscope and once the target cell is identified, the automated release needle pierces through the array and dislodges an individual microraft with the attached cell. The released microraft-attached cell is transferred to a secondary vessel by the magnetic wand for downstream analysis. For downstream WGA REPLI-gSingle Cell and for downstream WTA REPLI-gWTA Single Cell extraction kits (QIAGEN) may be further used. We view this simple, cost-effective, easy-to-use technique as a promising avenue for advancing our understanding of EBV involvement in periodontitis.

### 7.4. DNA Hybridization Arrays

The array basic lies in the immobilization of target-specific sequences (probes) on a solid matrix (nylon membranes, glass microscope slides, silicon or ceramic chips). These probe matrixes are then hybridized with labeled copies of nucleic acids from biological samples (targets). Probe-target hybridization is usually detected and quantified by the detection of the fluorophore-, silver- or chemiluminescence-labeled targets. The greater the amount of labeled target greater the output signal is. There are four array formats: macroarrays, microarrays, high-density oligonucleotide arrays (Gene Chips) and microelectronic arrays. All hybridization arrays include four steps: 1. DNA/RNA is isolated from samples; 2. an array with many target-specific probes is purchased or constructed; 3. labeled targets are generated from the sample DNA/RNA and 4. the targets are hybridized to the probes and the relative signals are measured. The hybridization array technology has been described in detail elsewhere [88]. Though this method has high sensitivity and specificity, it is confined to species for which probes are available [89].

The majority of the studies utilizing this array in periodontal health and disease applied it for the detection of bacterial species in mixed populations of SbgP samples (see the reviews from [89,90] and respective cited literature). To the best of our knowledge, hybridization arrays have not yet been employed for the analysis of EBV in periodontitis patients. Otherwise, recent studies report on clinical performance of the commercial multiplex PCR DNA microarray Clart Entherpex kit for detection of EBV, HCMV and HHV-6 in whole blood samples of hematopoietic stem cell-transplanted patients [91] or for simultaneous qualitative detection and identification of EBV, herpes simplex virus type 1 (HSV-1), HSV-2, HCMV, HHV-6, HHV-7 and HHV-8 in whole saliva samples of HIV-infected patients to establish a definitive diagnosis of OHL [92]. On the other side, lately, Wu et al. used the publically available periodontitis-related microarray data and observed an increased number of PCs in periodontitis-affected tissues versus those of healthy tissues [93].

### 7.5. Sequencing Techniques

High-throughput genomic sequencing techniques are the latest developments in oral and medical microbiology. Recent advances in NGS platforms, including the Roche/454 FLX genome sequencer, the ABI SOLiD system and the Illumina genome analyzer, empower the direct sequencing of EBV genomes in clinical tumors in a time- and cost-effective manner [94]. In the same manner, the WGS of periodontal tissues may be important in understanding the nature of the pathogenic EBV in vivo and its causative role in periodontitis.

For a detailed description of sequencing technologies, the readers are referred to reviews by [95,96,97]. As a simplified overview of the sequencing workflow, the gDNA can be extracted from the periodontal biopsies, subjected to library construction, then to short-read DNA sequencing using a genome analyzer and finalized by bioinformatics data analysis. The reduced costs of DNA sequencing democratized its usage in biology and medicine, however, it has brought also new challenges, such as NGS data analysis and interpretation of genetic data for a clinical utility [98].

RNAseq is another sequencing approach that utilizes NGS technology to study the entire transcriptome at high qualitative and quantitative levels. Unlike DNAseq, the RNAseq requires isolated RNA to be first reverse-transcribed into cDNA before sequencing. Understanding the transcriptome and the stepwise/sequential expression of viral genes are essential for interpreting EBV’s infectious life cycle associated with the disease progression.

RNAseq was used to detect the expression of not only latent but also lytic EBV genes in Burkitt’s lymphoma cells [99], gastric cancer cells [100] and nasopharyngeal carcinoma tissues [101]. Interestingly, the transcriptome sequencing of gingival biopsies from chronic periodontitis patients revealed that one of the top three upregulated genes was the complement receptor type 2, the EBV receptor of human B cells [102]. Undoubtedly, RNAseq is a valuable tool for understanding transcriptomic dynamics during the analysis of samples allowing comparison between diseased and healthy tissues, and categorization of disease states [103].

## 8. Moving Forward

Most of the PCR-based, immunohistochemistry, in situ hybridization, immunophenotyping methods described in this review were used more for research purposes to answer the specific scientific question of the etiopathogenic role EBV plays in periodontitis rather than for use in the diagnosis of a specific disease by detection of EBV. These methods were tools to accomplish what has been done in the area. On the horizon are futuristic approaches, which likely will improve further research. As the scientific questions are answered and the techniques improve, then it is forthcoming that many of these methods will be used as diagnostic tools.

## Figures and Tables

**Figure 1 microorganisms-09-00072-f001:**
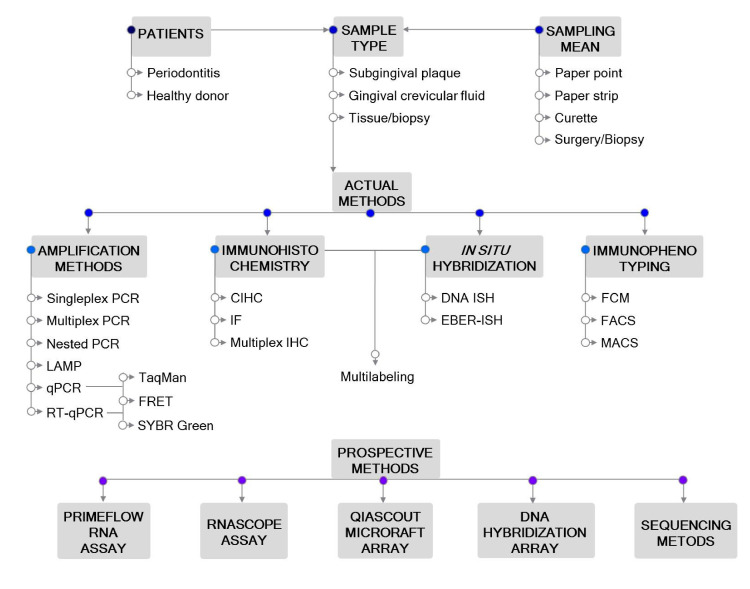
Outline of the methodological approaches for Epstein–Barr virus (EBV) detection in periodontitis. Actual and prospective methods are listed. Abbreviations: CIHC, chromogenic immunohistochemistry; EBER—EBV-encoded RNA; FACS—fluorescence-activated cell sorting; FCM—flow cytometry; FRET—fluorescence resonance energy transfer; IF—immunofluorescent detection; IHC—immunohistochemistry; ISH—*in situ* hybridization; LAMP—loop-mediated isothermal amplification; MACS—magnetic-activated cell sorting; PCR—polymerase chain reaction; qPCR—real-time quantitative PCR; RT-qPCR—reverse transcription qPCR.

**Figure 2 microorganisms-09-00072-f002:**
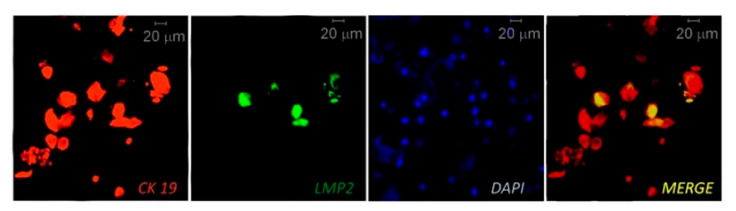
Immunofluorescent (IF) costaining of CK19 (junctional epithelial cell marker cytokeratin 19) and LMP2 (EBV latent membrane protein 2) to detect EBV-infected epithelial cells in samples taken from a periodontitis patient. The cell nuclei are counterstained with DAPI. Reprinted from [27].

**Figure 3 microorganisms-09-00072-f003:**
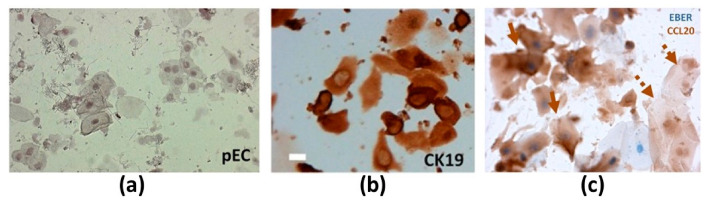
(**a**) EBV-encoded RNA in situ hybridization (EBER-ISH) of periodontal epithelial cells (pECs) and (**b**) chromogenic immunohistochemistry (CIHC) for CK19 (cytokeratin 19; junctional epithelial cell marker) immunostaining of periodontitis samples to reveal EBV-infected ECs. The size bar represents 15 µm. (**c**) EBER-ISH coupled with CIHC of CCL20 (chemokine ligand 20) to show the production of the inflammatory chemokine CCL20 by EBV-infected pECs in periodontitis patients. EBV-infected (solid arrows) and EBV^-^ pECs (dotted arrows) are presented. Reprinted from [27].

**Figure 4 microorganisms-09-00072-f004:**
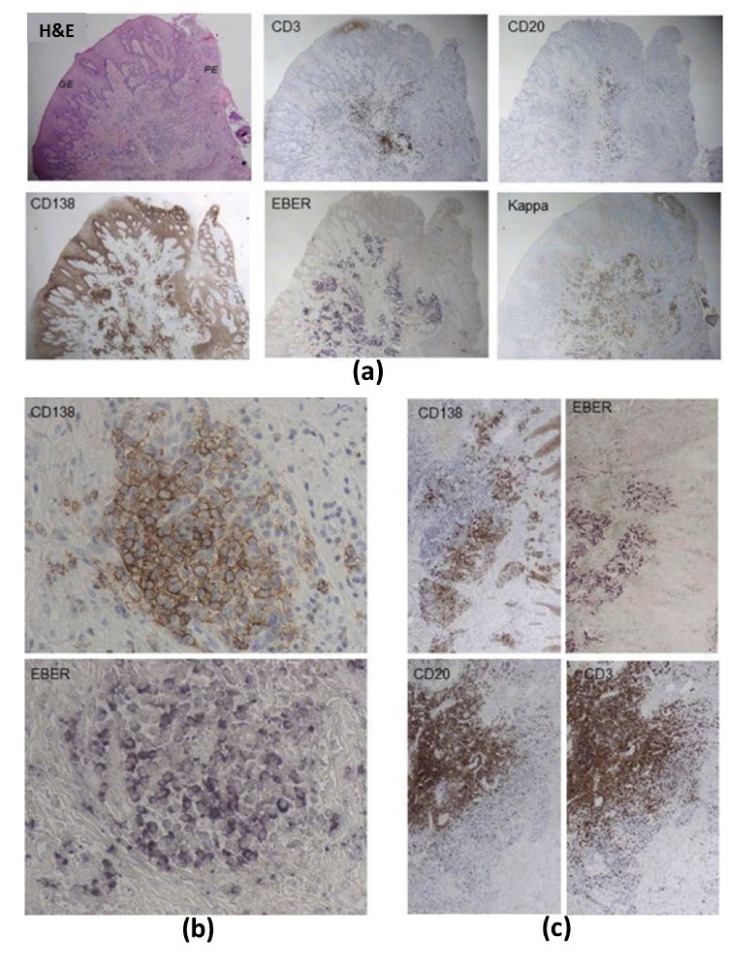
Multiplex chromogenic immunohistochemistry (CIHC) for the detection of T cells (CD3 (cluster of differentiation) marker), B cells (CD20), plasma cells (PCs; CD138) and antibody-producing cells (cytoplasmic kappa light chain) coupled with EBV-encoded RNA in situ hybridization (EBER-ISH) in the gingival tissue specimens of periodontitis patients. Panel (**a**) shows hematoxylin and eosin (H&E) overlapping between CD3, CD20, CD138, EBER and Kappa stainings on a specimen from a periodontitis patient (magnification ×4). Panel (**b**) illustrates a cluster of EBV-infected PCs showing colocalization between EBER and CD138 (magnification ×100). Panel (**c**) is indicative of colocalization between EBV-infected cells (EBER), PCs (CD138), B cells (CD20) and T cells (CD3) (magnification ×10). Reprinted from [62] with permission from Elsevier.

**Table 1 microorganisms-09-00072-t001:** Characteristics of the studies using nucleic acid amplification-based detection methods.

Study	Periodontitis Type	Sample Size	Sample Type	Sampling Type	Amplification Type	Target	EBV Occurrence		Main Findings
[38]	ApP	40 ApP	PApT	Curette	Nested PCR (DNA) RT (RNA) + Nested PCR (cDNA)	BamHI W (DNA) EBNA2 (RNA)	29 ApP DNA	20 ApP mRNA	EBV infection is a frequent event in ApP.
		40 HC	PT	NA			1 HC DNA	1 HC mRNA	
[37]	CP	40 CP: 40 SS + 40 DS	SbgP	Curette	Nested PCR (DNA) AGE	EBNA2	4 SS + 29 DS		Significant association of EBV1 and CP. Association between EBV1 and periodontopathic bacteria.
		40 HC					1 HC		
[31]	AgP, CP	20 AgP	SbgP	Curette	PCR (DNA) AGE	EBNA2	9 AgP		Significantly higher prevalence of EBV1 in AgP and CP subjects compared to HCs.
		20 CP					5 CP		
		20 HC					0 HC		
[33]	AgP, CP	10 patients: 25 AgPS + 25 CPS 25 HS	SbgP	Curette	Multiplex PCR (DNA) AGE	LMP2	8 AgPS + 8 CPS 2 HS		Significant association of EBV with CP and AgP.
[39]	CP	85 CP: 85 SS + 85 DS	SbgP	Paper point	Nested PCR (DNA) AGE	EBNA2	41 SS + 56 DS		More frequent detection of EBV DNA in patients with DS than in those with SS or HCs. EBV DNA may serve as a pathogenic factor leading to CP.
		20 HC: 40 HS					18 HS		
[56]	AgP	65 AgP	SbgP	Paper point	FRET qPCR (DNA)	BRLF1	7 AgP		No association between EBV and AgP.
		65 HC					9 HC		
[34]	AgP	15 AgP	SbgP, IDPT	Curette	Multiplex PCR (DNA) AGE	LMP2	10 AgP SbgP 11 AgP IDPT		Significant prevalence of EBV in AgP compared to HCs.
		15 HC					1 HC SbgP 0 HC IDPT		
[27]	CP	6 CP: 6 SS + 6 DS	SbgP	Curette	SYBR Green RT-qPCR (RNA)	EBNA1, EBNA2, LMP1, LMP2, BZLF1	6 SS + 6 DS: EBNA1 > EBNA2 ≥ LMP1 ≥ LMP2 ≥ BZLF1		EBV-specific latent (LMP1, LMP2, EBNA1 and EBNA2) and lytic (BZLF1) transcripts detected in all PP but not PalEC samples of CP patients.
		3 CP: 3 PalS	PalECs				3 PalS: EBNA1 = EBNA2 = LMP1 = LMP2 = BZLF1 = 0		
		10 HC: 10 HS	GS			EBNA1	DS > SS > HS		EBNA1 transcripts detected 36- and 5-fold higher in DS and SS, respectively, compared to HS.
[40]	ApP	100 ApP	PApT	Curette	Nested PCR (DNA) PAGE	EBNA2	76 ApP		Significant occurrence of EBV1 genotype in periapical lesions than in healthy pulps.
		25 HC	PT	Endodontic file			6 HC		
[32]	CP	100 CP	SbgP	Curette	PCR (DNA) AGE	LMP2	21 CP		Significantly higher levels of EBV in CP as compared to the healthy periodontium.
		100 HC					6 HC		
[28]	AgP	15 AgP	SbgP	Curette	Hotstart PCR (DNA) AGE	NA	6 AgP		EBV occurrence comparable among AgP and HC groups.
		15 HC					1 HC		
[57]	CP	60 CP	Tissue	Surgery	TaqMan qPCR (DNA)	NA	DS > SS		Observed EBV in tissue samples from deep and shallow PPs. Quantification of EBV is high in periodontal tissue samples of severe CP.
[58]	CP	25 CP: 25 SS + 25 DS	SbgP	Paper point	SYBR Green qPCR (DNA)	BNRF1	10 SS + 20 DS		Significantly high EBV DNA in DS than in SS of CP patients and HS of HCs. Association between EBV DNA, *P. gingivalis* and CP.
		13 HC: 26 HS					13 HS		
[35]	CP	40 CP	GCF	Paper strip	Multiplex PCR (DNA) AGE	LMP2	25 CP		Significantly higher prevalence of EBV in GCF of CP patients than in HCs. Strong association between EBV and CP.
		20 HC					2 HC		
[36]	CP (MiP, MdP, SvP)	100 MiP + 100 MdP + 100 SvP	SbgP	Curette, paper point	Multiplex PCR (DNA) AGE	LMP2	25 MiP + 20 MdP + 47 SvP		Significant association between EBV and CP, and the severity of the disease.
		300 HC					0 HC		
[54]	AgP	17 AgP	SbgP	Paper point	LAMP (DNA) AGE + TA	BamHI W	64.7% AgP		No significant association between EBV1 and AgP. Highest risk of AgP when *A. actinomycetemcomitans* and EBV1/HCMV are together.
		17 HC					47.1% HC		
[59]	GAP	165 GAP: 165 AS + 165 n-AS	SbgP	Paper point	qPCR (DNA) AGE	EBNA1	23 AS + NA n-AS		EBV association with *A. actinomycetemcomitans*. Although the presence of EBV (herpesvirus in general) is not necessary for the progression of GAP, it can facilitate it, possibly by promoting pathogenicity and virulence of periodontopathic bacteria in a virus and bacterial species-dependent manner.
[60]	AgP, CP	18 AgP +12 CP	SbgP	Curette	TaqMan qPCR (DNA)	NA	19 (AgP + CP)		Significant presence of EBV in periodontitis sites as compared to healthy sites. Positive correlation of EBV with *P. gingivalis* and *T. forsythia*.
		30 HC					3 HC		
[61]	AgP, ApP	22 AgP + 3 ApP	SbgP	Paper point	TaqMan qPCR (DNA)	EBNA1	16 AgP + 3 ApP		Prevalence and copy number of EBV significantly higher in periodontitis patients than in healthy controls.
		25 HC					4 HC		
[62]	CP (MdP, SvP)	20 patients: 9 MdP + 11 SvP	SbgP	Curette	TaqMan EBV R-GENE qPCR (DNA)	BXLF1	0–9861.14 × 10^2^ copies/µg		Different levels of EBV occurrence in CP patients.
[41]	AgP, CP	57 AgP	Tissue	Surgery	Nested PCR (DNA) AGE SYBER Green qPCR (DNA)	EBNA2 BALF5	25 AgP	4.41–7.01 log_10_ copies/g AgP	Significant occurrence of EBV in the AgP and CP groups compared to the HC. Significant association between EBV load and periodontitis.
		59 Cp					28 CP	5.06–7.31 log_10_ copies/g CP	
		43 HC					5 HC	4.57–5.21 log_10_ copies/g HC	

Abbreviations: Periodontitis type: AgP—aggressive periodontitis; ApP—apical periodontitis; CP—chronic periodontitis; HC—healthy control; GAP—generalized aggressive periodontitis; MdP—moderate periodontitis; MiP—mild periodontitis; SvP—severe periodontitis. Sample type: GCF—gingival crevicular(/periodontal pocket) fluid; GS—gingival sulcus; IDPT—interdental papilla tissue; PApT—periapical tissue; PalECs—palatal epithelial cells; PP—periodontal pocket; PT—pulp tissue; SbgP—subgingival plaque. Site type: AgPS—aggressive periodontitis site; AS—active site; CPS—chronic periodontitis site; DS—deep site; HS—healthy site; n-AS—non-active site; PalS—palatal site; SS—shallow site. Amplification type: FRET—fluorescence resonance energy transfer; PCR—polymerase chain reaction: qPCR—real-time quantitative PCR; RT—reverse transcription reaction; RT-qPCR—reverse transcription qPCR. Amplicon detection type: AGE—agarose gel electrophoresis; PAGE—polyacrylamide gel electrophoresis; TA—turbidity assay. NA—not available.

**Table 2 microorganisms-09-00072-t002:** Characteristics of the studies using tissue-based detection methods.

Study	Periodontitis Type	Sample Size		Sample Type	Sampling Type	Tissue-Based Detection Type		Target		EBV Occurrence		Main Findings
[66]	ApP	35 ApP		Apical lesion	Teeth extraction	CIHC		LMP1		11 ApP		EBV occurrence in about 31% of ApP samples.
[67]	ApP	20 ApP		Apical lesion	Submarginal incision	EBER-ISH		EBER		0 ApP		No signs of cells harboring EBV in 20 apical samples analyzed by EBER-ISH.
[39]	CP	41 SS + 56 DS		Gingival tissue	Flap surgery	EBER-ISH + CIHC		EBER CD19		EBER^+^ CD19^+^		Numerous CD19^+^ B cells infiltrated in the connective tissue subjacent to the gingival epithelium; numerous cells in the same location were EBER^+^.
[27]	CP	3 CP: 3 PP 3 CP: 3 PalS		SbgP PalECs	Curette, cytospin cuvette	IF costaining		LMP1, LMP2 CK19		3 PP 0 PalS		Around 32% of the CK19^+^ epithelial cells infected with EBV (LMP2^+^).
						EBER-ISH + CIHC		EBER CK19		EBER^+^, CK19^+^ PP EBER^-^, CK19^+^ PalS		EBER^+^ periodontal epithelial cells (pECs) were detected only in PP samples.
		20 CP: 20 SS + 20 DS 10 HC: 10 HS		SbgP GS	Curette, cytospin cuvette	EBER-ISH + CIHC		EBER CK19		DS > SS > HS		Frequency of EBV^+^ pECs higher in deep pockets than in shallow pockets and healthy sites. A positive correlation between EBV infection and disease severity.
[68]	PApP	9 PApP	6 PApP	PApG	Endodontic surgery	EBER-ISH	CIHC	EBER	LMP1	6 PApP	6PApP	EBER detected in the cytoplasm and nuclei of B cells and plasma cells (PC) in 66.7% of PApGs, but not in healthy gingival tissues. All EBER^+^ PApGs positive for LMP1. LMP-1-expressing cells localized in the same areas as EBER-expressing cells.
		5 HC	5 HC	Gingival tissue	Teeth extraction					0 HC	0 HC	
[62]	CP (SvP)	5 SvP		Gingival tissue	Surgery	EBER-ISH +		EBER		EBER^+^		Numerous EBV-infected cells, mostly overlapping with CD138^+^ PCs. EBV-infected PCs formed high-density clusters along the periodontal epithelium associated with CD3^+^ T cells and CD20^+^ B cells.
						Multiplex CIHC		CD3, CD20, CD138, Kappa		CD3^+^, CD20^+^, CD138^+^, Kappa^+^		

Abbreviations: Periodontitis type: ApP—apical periodontitis; CP—chronic periodontitis; PApP—periapical periodontitis; HC—healthy control; SvP—severe periodontitis. Sample type: GS—gingival sulcus; PApG—periapical granuloma; PalECs—palatal epithelial cells; SbgP—subgingival plaque. Site type: DS—deep site; HS—healthy site; PalS—palatal site; PP—periodontal pocket; SS—shallow site. Tissue-based analysis type: CIHC—chromogenic immunohistochemistry; EBER-ISH—EBV-encoded RNA in situ hybridization; IF—immunofluorescent staining.

## Data Availability

No new data were created or analyzed in this study. Data sharing is not applicable to this article.

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
