# Peer review of "Detection of Epstein–Barr Virus in Periodontitis: A Review of Methodological Approaches"

_microorganisms, 2020, doi:10.3390/microorganisms9010072_

Round 1

Reviewer 1 Report

This paper is a very well written, detailed summary of the techniques available to detect EBV infection in periodontitis – both those in use and those that look promising.  The wide variety of techniques presented will be helpful to those who need to detect EBV infection, as well as co-infections by other viruses and bacteria, in a variety of tissues and fluids that are often available in only small amounts.While there is a recent paper that reviews many of the techniques of EBV detection, this paper describes (very well) a somewhat different set, and uniquely discusses them in the context of periodontitis. 

Main comments:

A table summarizing tissue-based (IHC/ISH) studies, similar to the PCR table, would be very helpful.

The in situ hybridization section should mention the new, sensitive, and relatively simple "RNAscope" RNA/DNA ISH.

The sequencing section should mention the potential of RNAseq to reveal the behavior of EBV in infected tissue.

Minor comments:

Line 18 omit "the":  "promote long-term chronic inflammatory disease."

Line 49:  use "hypotheses", not "hypothesis"

Line 69:  recommend inserting "and,":  "stomach glands, and, occasionally, T cells, . . ."

Line 72:  recommend replacing ";" with "and"

Line 77:  recommend deleting "EBV is linked to a higher risk of development of certain autoimmune diseases, namely,”

Lines 134-137:  Salivary EBV load, especially, may be very relevant to periodontitis.  Please rewrite to clarify why you consider salivary EBV load less relevant to periodontitis than the parameters you have chosen to focus on.

To be fixed:  Page breaks in Table 1 divide data from a single study.

Line 273:  Please also comment on the studies in references 50 and 33, which come to opposite conclusions regarding the prevalence of EBV in AgP, but test for the lytic BRLF1 protein in one case and the latent LMP2 protein in the other – suggesting that EBV is latent in AgP.

Lines 304/305:  Please clarify this sentence by being more specific.

Line 364:  Recommend replacing "On the other hand," which suggests contrasting information, with "In addition"

Line 400:  Need an "and" for this list; recommend "and analyze . . . "

Line 403:  replace "has" with "have":  Antibodies (mAb) that have a specific . . ."

Lines 418/419:  Please rewrite and clarify the relationship between the existence of EBV-specific antibodies for FCM and the processing of biopsy material.

Line 433: omit "these"

The last three sentences of the "Moving forward" section should be more specific, or perhaps they could be condensed to a single sentence.

Author Response

This paper is a very well written, detailed summary of the techniques available to detect EBV infection in periodontitis – both those in use and those that look promising. The wide variety of techniques presented will be helpful to those who need to detect EBV infection, as well as co-infections by other viruses and bacteria, in a variety of tissues and fluids that are often available in only small amounts. While there is a recent paper that reviews many of the techniques of EBV detection, this paper describes (very well) a somewhat different set, and uniquely discusses them in the context of periodontitis.

We would like to thank the reviewer for the positive feedback and the kind words about our review. We greatly appreciate the time and energy the reviewer spent on our behalf. We found your thoughtful comments extremely helpful and have revised them accordingly.

Main comments:

Comment: A table summarizing tissue-based (IHC/ISH) studies, similar to the PCR table, would be very helpful.

Answer: We agree with the reviewer’s assessment and added Table 2 presenting the tissuebased detection methods.

Comment: The in situ hybridization section should mention the new, sensitive, and relatively simple "RNAscope" RNA/DNA ISH.

Answer: We thank the reviewer for pointing out this interesting technique. Considering RNAscope was not employed in periodontitis-EBV research, we have added the method in the Advanced prospective methods section as subsection 7.2 (L514-532). RNAscope was also added in Figure 1 of the revised version.

Comment: The sequencing section should mention the potential of RNAseq to reveal the behavior of EBV in infected tissue.

Answer: Thank you for the suggestion. We have added a section regarding RNAseq in L586597.

Minor comments:

Comment: Line 18 omit "the": "promote long-term chronic inflammatory disease." Answer: Thank you for the correction. “The” is deleted now.

Comment: Line 49: use "hypotheses", not "hypothesis"

Answer: Corrected.

Comment: Line 69: recommend inserting "and,": "stomach glands, and, occasionally, T cells, . . ."

Answer: “and,” is inserted.

Comment: Line 72: recommend replacing ";" with "and"

Answer: Replaced.

Comment: Line 77: recommend deleting "EBV is linked to a higher risk of development of certain autoimmune diseases, namely,”

Answer: Deleted. The sentence now reads as: “Furthermore, a recent study suggested the association of EBV with 7 different autoimmune diseases – multiple sclerosis, rheumatoid arthritis, inflammatory bowel disease, type 1 diabetes, juvenile idiopathic arthritis and celiac disease [13].”

Comment: Lines 134-137: Salivary EBV load, especially, may be very relevant to periodontitis. Please rewrite to clarify why you consider salivary EBV load less relevant to periodontitis than the parameters you have chosen to focus on. Answer: We strongly agree with the reviewer that salivary EBV load detection may be very relevant to periodontitis and is an important consideration, as illustrated by our own work (Olivieri et al., 2020). However, because saliva is one of the main reservoirs and the vehicle for EBV transmission, EBV can be commonly detected also in the saliva of healthy individuals. Thus, we excluded saliva from our discussion. We clarified this point in L140143: “Though salivary EBV load may be very relevant to periodontitis, in this review, the saliva as a sample and salivary EBV load detection methods are not discussed either, because EBV is ubiquitous in humans, it is transmitted through saliva and EBV DNA is commonly detected also in the saliva from healthy adults [22].”

Comment: To be fixed: Page breaks in Table 1 divide data from a single study. Answer: In our version of word document Table 1 is fixed now.

Comment: Line 273: Please also comment on the studies in references 50 and 33, which come to opposite conclusions regarding the prevalence of EBV in AgP, but test for the lytic BRLF1 protein in one case and the latent LMP2 protein in the other – suggesting that EBV is latent in AgP.

Answer: We cannot surely state that EBV is latent in AgP, e.g., the study from reference 54 observed no significant association between EBV and AgP testing for BamHI W promoter that is active during latent infection.

Comment: Lines 304/305: Please clarify this sentence by being more specific. Answer: We have clarified this sentence. Now it reads as follows: “Besides, because the evaluation of the staining intensity of IHC is subjective, the ambiguity in the evaluation of the results and inter- or intraobserver variability may be problematic [65].”

Comment: Line 364: Recommend replacing "On the other hand," which suggests contrasting information, with "In addition"

Answer: Thank you for the correction.

Comment: Line 400: Need an "and" for this list; recommend "and analyze . . . " Answer: Added.

Comment: Line 403: replace "has" with "have": Antibodies (mAb) that have a specific . . ."

Answer: Corrected.

Comment: Lines 418/419: Please rewrite and clarify the relationship between the existence of EBV-specific antibodies for FCM and the processing of biopsy material. Answer: We meant that when processing biopsy material to get single-cell suspensions, the enzymes may affect the viral protein of interest (personal observation), so the detection of EBV-encoded RNAs may be more reliable. However, we do not want to emphasize this point because it is just a preliminary observation, so we deleted that phrase. Now the sentence reads as: “In practice, reliable EBV-specific labeled antibodies for FCM are somewhat nonexistent, so the FCS is usually combined with EBER-ISH.”

Comment: Line 433: omit "these"

Answer: Deleted.

Comment: The last three sentences of the "Moving forward" section should be more specific, or perhaps they could be condensed to a single sentence.

Answer: We think that this section is rather specific, and condensing it to a single sentence will be too short to be considered as a section.

Reviewer 2 Report

In this review manuscript entitled "Detection of Epstein-Barr virus in periodontitis: A review of methodological approaches", Tonoyan and colleagues summarize multiple methods to detect EBV in human samples especially trying to focus on periodontitis.

Methodology itself is described comprehensively and correctly, however most parts are focused on methods applicable to (the reviewer thinks "being applied from") more general detection of EBV. Only sampling means are closely related to periodontitis but not specific for EBV detection. Overall, this manuscript is well-written as methodology review for molecular technique detecting EBV, however neither provides significant advance, fills the gap, nor informs future direction which are generally required for the review article.

Author Response

In this review manuscript entitled "Detection of Epstein-Barr virus in periodontitis: A review of methodological approaches", Tonoyan and colleagues summarize multiple methods to detect EBV in human samples especially trying to focus on periodontitis. Methodology itself is described comprehensively and correctly, however most parts are focused on methods applicable to (the reviewer thinks "being applied from") more general detection of EBV. Only sampling means are closely related to periodontitis but not specific for EBV detection. Overall, this manuscript is well-written as methodology review for molecular technique detecting EBV, however neither provides significant advance, fills the gap, nor informs future direction which are generally required for the review article.

We would like to thank the reviewer for the careful and thorough reading of our review, and positive comments on the presentation quality of our manuscript.

Comment: Methodology itself is described comprehensively and correctly, however most parts are focused on methods applicable to (the reviewer thinks "being applied from") more general detection of EBV. Only sampling means are closely related to periodontitis but not specific for EBV detection.

Answer: Because EBV has emerged as an etiopathogenic factor for periodontitis relatively recently, no gold standard and specific methods list yet exists to put EBV directly in a context of periodontitis, thus, the techniques separately used in both periodontitis and EBV research are “borrowed” to “serve the common good”. The ultimate goal is to confirm the involvement of EBV in periodontitis and use EBV detection as a tool to diagnose and prevent periodontitis.

Comment: Overall, this manuscript is well-written as methodology review for molecular technique detecting EBV, however neither provides significant advance, fills the gap, nor informs future direction which are generally required for the review article.

Answer: We respectfully disagree with the reviewer on the statement that our review does not provide significant advance. We believe our manuscript provides an advance in the field because it summarizes the techniques and studies which implemented general scientific and technological advancements to decipher the role of EBV in periodontitis. Notably, it provides researchers (specifically in dentistry) who are not all familiar with virology and, in particular, with EBV, new means to broaden the scope of their investigations. Confirmed EBV infection in periodontitis may promote the development of novel pharmacological therapies. Additionally, although this review represents a practical guide for the periodontitis model, it may also pave the way for the understanding of other inflammatory infectious diseases associated with EBV infection, such as multiple sclerosis, rheumatoid arthritis, Sjogren's syndrome, etc. To emphasize the contribution of our review to overall research advancement, we have added a sentence in L118-120:
“Although this review represents a practical guide for the periodontitis model, it may also pave the way for the understanding of other inflammatory infectious diseases associated with EBV infection.”

We think that this manuscript is pointing out the future directions presenting prospective methods that were never applied specifically in the periodontitis-EBV research axis but hold great potential for application. As it was mentioned by reviewer 3, we point out “how these tools may be used beyond research studies, but to utilization in disease (periodontitis) detection”.

We also think that the manuscript fills the gap delineating the interdisciplinary links between odontology, microbiology, virology and immunology to advance the knowledge of periodontitis and to solve a complex societal problem such as the prevention and treatment of this disease. We respect the judgment and expertise of the reviewer, but we hope the clarifications provided above can earn a re-evaluation of our work from the reviewer.

Reviewer 3 Report

Summary:

Periodontitis is a risk factor for many other diseases. Periodontal disease is linked to pathogens, such as Epstein-Barr virus. This article reviews methods for detecting EBV specifically within periodontitis, including both procedures for obtaining samples as well as the detection methods.

Broad comments

This is a detailed review article that provides a substantial amount of background information in both periodontitis and EBV. The techniques are well described and references are provided for the use of the technique with EBV. The large table of findings on EBV in various types of periodontitis using PCR-methods nicely summarizes the literature. Finally, newer methodologies that may be applied to the detection of EBV are discussed. The authors also point out how these tools may be used beyond research studies, but to utilization in disease detection.

Author Response

Summary: Periodontitis is a risk factor for many other diseases. Periodontal disease is linked to pathogens, such as Epstein-Barr virus. This article reviews methods for detecting EBV specifically within periodontitis, including both procedures for obtaining samples as well as the detection methods.

We thank the reviewer for the careful review and accurate summary of our work.

Broad comments: This is a detailed review article that provides a substantial amount of background information in both periodontitis and EBV. The techniques are well described and references are provided for the use of the technique with EBV. The large table of findings on EBV in various types of periodontitis using PCR-methods nicely summarizes the literature. Finally, newer methodologies that may be applied to the detection of EBV are discussed. The authors also point out how these tools may be used beyond research studies, but to utilization in disease detection.

We would like to thank the reviewer for taking the precious time and effort necessary to review our manuscript. We are grateful for the reviewer’s positive feedback and encouraging words.

Round 2

Reviewer 2 Report

In the revised version for review manuscript entitled "Detection of Epstein-Barr virus in periodontitis: A review of methodological approaches", Tonoyan and colleagues added some sentences but the reviewer does not think the manuscript is sufficiently improved as review article.  

Below lists are the reviewer's opinion to add for improving the manuscript to "provide researchers (specifically in dentistry) who are not all familiar with virology and, in particular, with EBV, new means to broaden the scope of their investigations", but these are not all.  

1.There is no explanation which state is relevant for (causing) EBV pathogenesis in periodontitis, lytic infection or latency. EBV can show pathogenesis in both phases. Without clearing this point, most part in the manuscript seems to aim detecting EBV latency.  If the methods mentioned in this review could have been clarifying it, have to add relating literatures to show the importance of this review. Without these clarification or usefulness of the method in detecting EBV in periodontitis, this review just introduces the methodology detecting EBV in any samples.

2.In the manuscript, authors explain why they don't mention about detecting EBV in the blood or saliva, however these seem essential by means of specifying contribution of EBV in periodontitis. Periodontitis is an inflammatory disease and difficult to avoid contamination of the blood or saliva during sampling. Also the blood is the origin of saliva. Without comparing EBV status (viral load or gene expression profile) among periodontitis sample, saliva and blood, how could we know EBV contribution in periodontitis specifically (more enriched, only reactivated in periodontitis sample or distinct status of latency [0-III] etc.)? 

3. In section 7, several advanced technologies are mentioned (overly) in detail. Also the authors list specific company's products. The reviewer does not think this section is essential (can be omitted) if just introduce their future plan to apply these specific technologies. Instead better to expand their view about difficulties in sampling or detecting EBV in periodontitis with the current technologies, then refer these advanced technologies while citing critical examples applicable to EBV-periodontitis research field.